# Electrospun Poly-L-Lactic Acid Scaffolds Surface-Modified via Reactive Magnetron Sputtering Using Different Mixing Ratios of Nitrogen and Xenon

**DOI:** 10.3390/polym15132969

**Published:** 2023-07-06

**Authors:** Pavel V. Maryin, Tuan-Hoang Tran, Anastasia A. Frolova, Mikhail A. Buldakov, Evgeny L. Choinzonov, Anna I. Kozelskaya, Sven Rutkowski, Sergei I. Tverdokhlebov

**Affiliations:** 1Weinberg Research Center, School of Nuclear Science & Engineering, National Research Tomsk Polytechnic University, 30 Lenin Avenue, 634050 Tomsk, Russia; pvm5@tpu.ru (P.V.M.); cungbinh9327@gmail.com (T.-H.T.); kozelskayaai@tpu.ru (A.I.K.); 2Cancer Research Institute of Tomsk National Research Medical Center of Russian Academy of Sciences, 5 Kooperativny Street, 634050 Tomsk, Russia; frolova_aa@onco.tnimc.ru (A.A.F.); buldakov@oncology.tomsk.ru (M.A.B.); choynzonov@tnimc.ru (E.L.C.)

**Keywords:** electrospun PLLA scaffold, reactive magnetron sputtering, working gas mixtures, nitrogen-containing titanium coating, proliferative activity, HOS cells

## Abstract

Controlled regeneration processes involving tissue growth using the surface and structure of scaffolds, are actively used in tissue engineering. Reactive magnetron sputtering is a versatile surface modification method of both metal and polymer substrates, as the properties of the formed coatings can be modified in a wide range by changing the process parameters. In magnetron sputtering, the working gas and its composition have an influence on the chemical composition and physical characteristics of the obtained coatings. However, there are no studies addressing the influence of the nitrogen/xenon gas mixture ratio in direct current magnetron sputtering on the deposition rate, physicochemical and in vitro properties of surface-modified biocompatible poly-L-lactic acid scaffolds. In this study, the application of mixtures of nitrogen and xenon in various ratios is demonstrated to modify the surface of non-woven poly-L-lactic acid scaffolds by direct current magnetron sputtering of a titanium target. It has been found that the magnetron sputtering parameters chosen do not negatively influence the morphology of the prepared scaffolds, but increase the hydrophilicity. Moreover, quantitative spectroscopic analysis results indicate that the formed coatings are primarily composed of titanium oxide and titanium oxynitride compounds and is dependent on the gas mixture ratio only to a certain extent. Atomic force microscopy investigations of the roughness of the fibers of the electrospun scaffolds and the thickness of the coatings formed on them show that the considerable variations observed in the intrinsic fiber reliefs are due to the formation of a fine layer on the fiber surfaces. The observed decrease in roughness after plasma modification is due to temperature and radiation effects of the plasma. In vitro experiments with human osteosarcoma cells show that the scaffolds investigated here have no cytotoxic effect on these cells. The cells adhere and proliferate well on each of the surface-modified electrospun scaffolds, with stimulation of cell differentiation in the osteogenic direction.

## 1. Introduction

In recent decades, tissue engineering has made extensive use of guided bone regeneration, in which bone growth is controlled based on the surface and structure of the implanted scaffold [1]. Bone growth is primarily determined by the material from which the scaffold is made [2]. However, the brittleness of ceramics traditionally used for bone regeneration limits the application range of these materials [3]. However, advances in medical materials science have solved this problem with the use of bioresorbable polymeric materials. The major benefits of these materials is their ability to decompose in the biological environment of the human body and are gradually replaced by re-growing functional tissues. Bioresorbable polymeric materials derived from natural resources (e.g., collagen) and synthetically produced polymeric materials (e.g., polycaprolactone, polylactic acid) are widely used for scaffold fabrication [4]. Nowadays, poly-L-lactic acid (PLLA) is applied in the fabrication of non-woven scaffolds because of its excellent physico-chemical and mechanical properties, which are commonly prepared by electrospinning [5]. The high strength and considerable relative elongation of scaffolds fabricated from PLLA by electrospinning allow them to be broadly applied in the area of biomedicine. However, the application of PLLA scaffolds is restricted because of considerable high hydrophobicity and low degree of fiber surface functionalization, preventing cell adherence as well as proliferative activity [6].

Among the most potential options to modify the PLLA scaffold surfaces and impart high free surface energy to them is the deposition of thin single-component metal and complex nitride, oxide and oxynitride coatings by physical vapor deposition (PVD) methods in a reactive gas atmosphere [7]. This method is based on the production of chemically active titanium ions created by bombarding a titanium target with the working gas molecules during magnetron sputtering. In this case, the chemical reaction of the interaction between titanium and nitrogen takes place not in the plasma of the magnetron discharge, but on the surface of the scaffolds (substrate) made of the polymer PLLA [8]. Plasma deposited nanoscale thick nitrogenous titanium oxide (TiNxOy) layers are now becoming more and more important for biomedicine and tissue engineering due to the fact that their physicochemical and biomedical properties can be controlled by the structure and chemical composition of these coatings [7]. Moreover, these TiNxOy coatings exhibit high biocompatibility, high corrosion resistance, good mechanical behavior [9], as well as antibacterial properties [10].

Various ion-plasma methods are currently used for the formation of thin titanium oxynitrides (TiO_x_N_y_) films. For example, processing methods in “cold” plasma of various discharges are widely used: corona, barrier, magnetron and radiofrequency [11]. Working gas-applying magnetron sputtering has become the utmost diverse process for the deposition of nitrogenous titanium oxide containing thin films on polymer substrates because the working gas consistence, the sputtered target, the applied current and some more variables can be widely changed [12]. As a fact, the type of the working gas utilized greatly impacts the atomization rate, particle velocity, ionization area, ionization potential of the working gas and the degree of the Penning effect [13], in consequence enabling the generation of coatings with tunable physico-chemical features. For this reason, the employment of working gas mixtures (one reactive and one inert gas) for the deposition of different TiN_x_O_y_ containing layers has become common [14]. For example, for enhancing the wettability properties of polystyrene coatings, Clément et al. performed magnetron sputtering in a mixture of nitrogen (N_2_)/argon (Ar) and oxygen (O_2_)/Ar, and their mixing ratio varied from 0% to 100% [15]. It was also shown that the ratio of gases in the chamber affects the metastable states of the particles and the electron temperature [16]. Previously, nitrogen oxide (NO)-containing calcium phosphate films has also been deposited on titanium based media using magnetron sputtering in the radio frequency (RF) mode, controlling the amount of NO-containing compounds in the coatings by increasing the N_2_ concentration in the N_2_/Ar gas mixture [17]. In a previous work, it was shown that the coatings obtained by direct-current (DC) magnetron sputtering using the mixtures N_2_ + Xe and N_2_ + Ar exhibited the best degree of substrate colonization by fibroblasts compared with coatings prepared with N_2_ only [18].

Moreover, magnetron sputtering with a working gas mixture enables the coating of complicated frameworks, like polymeric bioresorbable scaffolds made of PLLA, which are vulnerable to thermal and radiation damage [11]. Through variations in the formulation and proportion of the working gases applied during the sputtering process, it is feasible to perform not merely a “gentle” modification of biodegradable non-woven scaffolds from PLLA, but also to influence the deposition efficiency as well as the physico-chemical characteristics of the generated thin films. However, there exist a lack of studies addressing the role of a N_2_/Xe working gas mixture in DC magnetron sputtering on the rate of deposit of NO-containing titanium films formed on biodegradable PLLA scaffolds.

Thus, this study intends to analyze the impact of the nitrogen and xenon gas mixing ratio during sputtering of NO-containing titanium thin films by direct current magnetron sputtering towards the physicochemical as well as the biomedical characteristics for PLLA scaffolds, which can be used for the surgical repair of bone damages and medical disorders and somatic injuries.

## 2. Materials and Methods

### 2.1. Electrospun Scaffold Fabrication

Electrospun poly-L-lactic acid (PLLA) fibrous scaffolds were prepared using a 3 wt.% electrospinning solution of PLLA (PURASORB^®®^PL-18, average molecular weight; 217,000–225,000 g/mol, Corbion Purac, Amsterdam, The Netherlands) in trichloromethane (CHCl3, 97 wt.%) (EKROS, Saint Petersburg, Russia) to study the effects of different nitrogen/xenon mixing ratios on physicochemical and biological properties. The scaffolds were prepared employing the below electrospinning criteria: —needle-collector distance: 110 mm, —collector Ø: 100 mm, —collector length: 210 mm, —collector rotation speed: 50 rpm, —flow rate: 4 mL/h, —spinneret speed: 10 mm/s, —voltage: +22 kV, —needle: 20 G. To remove remaining CHCl_3_, the scaffolds were placed in a vacuum oven (VTSH-K24-250, Aktan, Moscow, Russia) at a pressure of 10^−2^ Pa at 100 °C for 10 h prior to surface-modification.

### 2.2. Scaffold Modification

Plasma alteration of the PLLA scaffold fiber surfaces via nitrogenous titanium coatings has been performed using reactive DC magnetron sputtering of titanium (Ti, 99.99% NPK OboronMetChim, Tomsk, Russia) in a nitrogen (N_2_, 99.999%, Metal Bureau, Tomsk, Russia) and xenon atmosphere (Xe, 99.9995%, PTK Kryogen, Aramil, Russia) at different gas mixing ratios: N_2_ 100% (N_2_:Xe = 100:0), N_2_ 25% + Xe 75% (N_2_:Xe = 75:25), N_2_ 50% + Xe 50% (N_2_:Xe = 50:50), N_2_ 75% + Xe 25% (N_2_:Xe = 25:75) and Xe 100% (N_2_:Xe = 0:100). A Katod-1M magnetron sputtering device (Juterma, Rostov-on-Don, USSR) was used for coating fabrication, which is described in more detail in reference [19]. The process conditions employed for the preparation of the coatings were as listed here: —titanium target-sample-distance: 33 mm, —magnetron chamber pressure: 3 × 10^−3^ Pa, —magnetron chamber working pressure: 0.8 Pa, and the area of the atomized target was 224 cm^2^. Each of the two working gases, nitrogen (N_2_) and the noble gas xenon (Xe) has been introduced to the magnetron sputtering working chamber in different mixing ratios, as mentioned above. In total, the modification procedure has been carried out for two minutes at a power of 88 W. For this purpose, this surface treatment has been executed using a rotary regime to minimize the destruction to the polymeric samples during plasma impact thusly: one minute of plasma treatment, with an intermission of three minutes followed by another one minute of plasma surface modification.

### 2.3. Scaffold Properties Investigations

The morphology of the prepared PLLA scaffold samples has been examined employing scanning electron microscopy (SEM) (Quanta 200 3D, FEI, Hillsboro, OR, USA). All SEM micrographs has been handled with ImageJ 1.53 software (National Institutes of Health, Bethesda, MD, USA). In order to estimate the mean fiber diameters of a scaffold sample, the diameter of not less than 100 fibers has been valued with use of at least five SEM micrographs taken at different locations on a scaffold. Pore areas and mean porosities were calculated from five SEM micrographs at different areas on the surfaces of the PLLA scaffolds using ImageJ 1.38. Each sample group included at least three samples.

Fiber topography and the thickness of the samples surface-modified with nitrogenous titanium layers on the surface of PLLA scaffold samples, as well as root mean square (RMS) roughness, were measured in semi-contact mode (in this mode, the tip periodically touches the sample surface [20] using an atomic force microscope (AFM, AFM-Raman, NT-MDT, Zelenograd, Russia). This device was equipped with an NSG01 tip (NT-MDT, Zelenograd, Russia), which operated with the typical force constant in the range of 1.45–15.1 N/m and a resonant frequency in the interval of 84 kHz–230 kHz. The samples were attached with a double-sided adhesive tape. Three random areas and three randomly chosen fibers of the sample surfaces were scanned to obtain the topography. Results obtained by AFM analysis have been handled with the software Gwyddion 2.62 (http://gwyddion.net, Brno, Czech Republic, accessed on 12 April 2023).

The chemical formulation of all coatings has been analyzed by means of XPS (X-ray photoelectron spectroscopy, Escalab 250Xi, Thermo Fisher Scientific Inc., Waltham, MA, USA) employing AlKα rays (with the energy of a photon: 1486.6 eV) having an energy resolution in total of 0.3 eV for all measurements. XPS Spectra have been recorded using the constant pass energy mode (50 eV for the survey spectra and 20 eV for the elemental core level spectra) with a beam spot of 650 µm in diameter. All measurements have been conducted at room temperature inside an UHV (ultra-high vacuum) having a pressure of 1 × 10^−9^ mbar (when the electron-ion compensation system was used, the partial pressure of argon equalled 1 × 10^−7^ mbar). The deconvolution spectra of N1s, C1s, O1s and Ti2p have been processed by using the Voigt function in the software OriginPro 9.0 (OriginLab Corporation, Northampton, MA, USA).

Tests on the wettability of the PLLA scaffold specimens have been conducted utilizing a drop shape analyzer (Easy Drop DSA-20, Krüss, Hamburg, Germany), employing the sessile drop method. To carry out the measurements, three droplets of glycerol with the volume of 3 µL were applied on the scaffold surfaces in air. For the assessment of the wetting dynamics, glycerol contact angles have been taken instantly and after one minute the droplets were applied to the specimens. Glycerol has been selected, because it has higher viscosity and surface tension in comparison to water, whereby the contact angles of all polymer scaffold samples could be determined [21]. With water wetted the prepared PLLA samples very effectively, as the water has been fully soaked up right after having come into contact with the PLLA scaffold surfaces. A similar behavior has been found using diiodomethane and formamide. There were three specimens in every PLLA scaffold sample group having a dimension of 5 × 40 mm^2^.

Human osteosarcoma (HOS) cells from the Russian collection of vertebrate cell ocultures of the Institute of Cytology of the Russian Academy of Sciences (Russian Academy of Sciences, St. Petersburg, Russia) was used as the cell line for the in vitro tests. Cells were incubated in Roswell Park Memorial Institute (RPMI) medium (Paneco, Moscow, Russia) with the addition of 10% inactivated calf serum (HyClone, Logan, UT, USA) and the following antibiotics: Penicillin—5000 units/mL and Streptomycin—5000 µg/mL (Paneco, Moscow, Russia). Prior to cell seeding, the scaffold samples with the size of 1 cm^2^ were completely soaked in the nutrient medium for 30 min and then moved into 24-well plates containing no nutrient solution. A 20 μL cell suspension containing 50,000 cells was applied to each sample with a hemocytometer (Hausser Scientific, Horsham, PA, USA) and allowed to stand for 15 min. This treatment has been performed to prevent leakage (“washout”) of cells when the samples with cells are immediately contacted with the liquid nutrient medium. In this way, adherence of the cells to the samples is increased, thus ensuring the equal amount of cells is present onto each sample. Subsequently, 200 μL of the RPMI medium was given into every well of the culture plate and incubated with 37 °C and 5% carbon dioxide (CO_2_). The total incubation time was 14 days until all tests were completed.

The cytotoxic effect of the samples towards the HOS cells has been assessed by MTT assays (3-(4,5-dimethylthiazol-2-yl)-2,5-diphenyltetrazolium bromide—MTT; Paneco, Moscow, Russia) assay. For this purpose, the samples were circular and had an area of 1 cm^2^. To do this, 5 mg/mL of MTT was added to each well with samples under investigation and incubated up to 2 h, until the color of the solution changed. After incubation, MTT has been taken away and DMSO (dimethylsulfoxide; Paneco, Moscow, Russia) has been given to every well. The optical density has been evaluated with the wavelength being 570 nm employing a Multiskan FC microplate photometer (Thermo Fisher Scientific, Waltham, MA, USA) (the higher the measured optical density, the higher the proliferative activity of the cells; with the measured result of the control samples was considered to be 100%). Measurements were carried out on days 1, 3, and 5 subsequent to cell seeding. Each MTT assay has been conducted in triplicate.

Cell adherence has been assessed via fluorescence staining of cells adhered on the examined samples. In order to do this, the supernatant from the wells was removed and the samples were washed twice with phosphate-buffered saline (PBS; Paneco, Moscow, Russia). Cells were subsequently fixed with a 2% formaldehyde solution (AppliChem GmbH, Darmstadt, Germany) at 4 °C temperature over 20 min. Following fixation, samples had been washed twice using a PBS solution. Next, the samples were stained using nuclear dye DAPI (2-(4-Amidinophenyl)-1H-indole-6-carboxamidine; Invitrogen, Waltham, MA, USA). Fluorescence micrographs of the stained cells were taken on days 1 and 5 after cell seeding. Each adhesion assay was performed in triplicate. Visualization was performed using an EVOS M7000 imaging system (Thermo Fisher Scientific, Waltham, MA, USA) equipped with a ×10 objective lens. Cell counts were performed in 10 different fields of view. Micrographs were taken on days 1 and 5 after cell seeding.

Differentiation at the stage of osteoblast formation was assessed by real-time polymerase chain reaction expression of the osteocalcin BGLAP and osteopanthin SPP1 genes on day 14 using. For this purpose, total RNA extraction was carried out by the column method using an RNeasy Mini Kit (Qiagen, Germantown, MD, USA). The extracted RNA was utilized as a matrix for cDNA synthesis by using a RevertAid kit (Thermo Fisher Scientific, Waltham, MA, USA) according to the manufacturer’s protocol. Gene expression was assessed by quantitative real-time PCR (qRT-PCR) using the TaqMan probe principle under the following conditions: 5 min at 95 °C; (10 s at 95 °C and 30 s at 60 °C) × 45 cycles; and 7 min at 72 °C. PCR amplification was performed in a programmable thermal cycler (AriaMx Real-Time PCR System, Agilent Technologies Inc., Santa Clara, CA, USA). Each qRT-PCR was performed in triplicate, with RNA quantity normalized to GAPDH content, and gene expression quantified according to the 2^−ΔΔCt^ method. The qRT-PCR primers used were as follows: for the bone gamma-carboxyglutamate protein (BGLAP), the direct primer 5′-TGCAGCCTTTGTGTCCAA-3′ and the reverse primer 5′-GCTCCCAGCCATTGATACA-3′; for the secreted phosphoprotein 1 (SPP1), the direct primer 5′-CCTGTGCCATACCAGTTAAACA-3′ and the reverse primer 5′-AGCATCTGGGTATTTGTTGTAAAG-3′; and for the glyceraldehyde 3-phosphate dehydrogenase (GAPDH), the direct primer 5′-GCCAGCC.

### 2.4. Statistical Evaluation

The statistical treatment of the acquired data has been performed with the software Statistica 7.0 (StatSoft, Tulsa, OK, USA). The majority of the data are expressed using the mean ± standard deviation. At *p* > 0.05, differences have been assumed to be statistically significant. One-way ANOVA test was used to determine the significance of the observed results for the sample groups.

Figure 1 highlights the schematic diagram of the experiments showing the procedure of PLLA scaffolds fabrication, their surface modification and the research techniques applied in this work.

## 3. Results and Discussion

### 3.1. Surface Morphology

Figure 2 shows micrographs obtained by scanning electron microscopy (SEM) at ×4000 and ×40,000 magnifications, the distribution of scaffold fiber diameter, and the histograms of the scaffold pore area of the unmodified and surface-modified PLLA scaffolds.

Morphological structure of PLLA scaffolds, regardless of the mixing ratio of the working gases, consists of a series of fibers that are chaotically interwoven. In this context, the fibers are predominantly cylindrical in shape, with negligible polydispersity and a normal distribution of the average fiber diameter in the range of (1.7 ± 0.4) μm (Figure 2). Deficiencies in form of molten fibers or burnouts on the fiber surface are not observed. High magnification SEM micrographs (insets in Figure 2), no defects are observed even on the fiber surfaces. It should be mentioned that the structure is preserved during surface modification without depending on the volumetric concentration of the working gases in the vacuum chamber. The mean pore areas and mean porosity are reliably consistent for all experimental scaffold groups and are independent of the mixing ratio of nitrogen and xenon (Figure 2, right).

### 3.2. Results of the Wettability Measurement

Findings from the wettability tests on the PLLA scaffold samples for the tested working gas mixture ratios after one minute of glycerol contact are shown in the Figure 3a.

Unmodified PLLA scaffolds fabricated by electrospinning have a wetting angle of 123° ± 6° and thus a hydrophobic surface [22]. Surface modification via plasma treatment in a 100% nitrogen (N_2_, N_2_ 100%) atmosphere decreases the value of the wetting angle to 82° ± 3°. When xenon (Xe) is added to the vacuum chamber, the wetting angles does not change linearly, and there is a dependence on the mixing ratio of nitrogen and xenon. Thus, the scaffold samples obtained with the mixing ratio N_2_ 50% + Xe 50%, is characterized by a minimum value of the wetting angle of 32° ± 1°. Moreover, no significant differences are found between the scaffold samples N_2_ 75% + Xe 25%, N_2_ 25% + Xe 75% and Xe 100%, but the wettability is reliably better than that of the control samples. The dependence of the wettability on the mixing ratio of the gases in the vacuum chamber is due to several parallel processes.

Firstly, different gas atmospheres allow the formation of different stoichiometric nitrogenous titanium coatings deposited onto the scaffold fibers (Figure 3 and Appendix A). Thus, for the mixing ratio N_2_ 50% + Xe 50%, the layers created onto the scaffold fibers are having the lowest coating thickness and exhibit the highest wettability. On the other hand, the N_2_ 100% samples exhibit the highest values for the contact angles. A similar effect was observed in studies in which the surface of medical devices was modified with different types of discharges by saturating their surface with nitrogen [23,24]. It is also known that wettability increases with increasing surface roughness [25,26].

Thicknesses of obtained thin nitrogen-containing titanium coatings on the surfaces of the PLLA scaffold samples are shown in Figure 3b. After surface modification of the scaffold fibers, significant changes in their relief are observed, which is due to the deposition of thin nitrogenous titanium coatings having a coating thickness dependent on the mixing ratio of N_2_ to Xe (Figure 3b). The highest coating thickness is observed for the pure Xe atmosphere, and the lowest for the N_2_ 50% + Xe 50% mixed gas atmosphere. This fact is due to plasma processes described below. It is known that the deposition rate of thin film coatings depends essentially on the working gas atmosphere composition in the magnetron sputtering chamber [27], and the ionization cross-section of the plasma-forming gas increases with the increase of its atomic number [27].

Since the value of the ionization cross-section of xenon is higher than that of nitrogen [27], the thickness of the coatings formed in an atmosphere of pure xenon is higher (Figure 3b). It is also worth noting the lower efficiency of the Penning effect for the mixture ratios of N_2_ to Xe, since their ionization potentials differ significantly, and the discharge power is quite lower [28]. However, for a more complex composition of a working gas, the interaction process is not linear and depends on the nitrogen and xenon concentrations (mixing ratios), the Penning effect, the temperature of the ions, the ion heating by wave attenuation and Coulomb scattering.

### 3.3. Chemical Composition of the PLLA Scaffolds

Figure 4 shows the high-resolution XPS spectra of carbon (C1s), titanium (Ti2p), nitrogen (N1s) and oxygen (O1s) of the fabricated samples. In addition, Appendix A presents the data for the elemental concentrations of titanium, carbon, oxygen and nitrogen determined during XPS analysis.

There are no peaks characterizing titanium and its compounds at the Ti2p spectrum obtained from the PLLA control samples (Figure 4a). After plasma treatment, a doublet of spin orbits is observed in the spectrum at the binding energies ~457.6 eV and ~464.2 eV, corresponding to the Ti2p3/2 and Ti2p1/2 electronic states, depending on the mixing ratio of the working gases. Moreover, a satellite characteristic of titanium oxides is observed at binding energies of ~471.6 eV [29] and is due to the transition of the valence electron to a higher energy level [30]. The binding energy of ~457.6 eV indicates the existence of TiN_y_ compounds, and the binding energy of ~464.2 eV indicates the presence of TiO_x_ bonds, suggesting the existence of complex Ti-O-N compounds within the composition of the coatings formed on the scaffold fibers [31]. Noteworthy is also the shift of the peak center at ~457.6 eV by ~0.3–0.4 eV toward higher binding energy, which is associated with the formation of surface defects because of plasma exposure [32]. Pure xenon plasma creates defects on the surface of the scaffold (vacancies, trapped charges, etc.) and reduces the kinetic energy of photoelectrons, leading to an increase in binding energy. When nitrogen is added to the vacuum chamber, the density of surface defects decreases and the PLLA scaffold sample “heals”, followed by decreasing binding energy [33].

The oxygen (O1s) spectrum of the PLLA control sample (Figure 4b) indicates two peaks at binding energies of ~533.2 and ~531.1 eV, respective to C=O and C-O bonds of the polymer. Subsequent to the surface treatment process, a reduction of the relative intensity for the C=O peak is observed at each gas mixing ratio, indicating a breakdown of the double bond. Moreover, subsequent the surface modification, the appearance of one intense peak around ~529.7 eV binding energy is noted, corresponding to Ti-O compounds, which corresponds to O^2−^ anion within a TiO_2_ lattice. A shift in the peak center at ~529.7 eV by ~0.3 eV toward higher binding energy is also observed during the modification of PLLA scaffolds in pure xenon atmosphere, which is due to the mechanisms described above for the Ti2p spectrum.

The C1S spectra (Figure 4c) obtained from the non-modified PLLA control sample show 3 major peaks in the 284.8 eV (C-H/C-C), 286.8 eV (C-OH/C-O-C) and 289.2 eV (C=O/O-C=O) regions, representing the most important compounds of the polymer PLLA [34]. Following plasma treatment, there is a significant decrease in the intensity of the peak at 289.2 eV independently from the working gas mixing ratio, due to destruction on PLLA by plasma, as described in reference [18].

No peaks characteristic of nitrogen compounds within the nitrogen (N1s) spectrum for the control sample (Figure 4d). Upon surface modification treatment, this spectrum exhibits a peak having the maximum around ~399.6 eV and a broad shoulder in the region of 402–400 eV, with a maximum at ~401.8 eV. The interpretation of the peak at ~399.6 eV is not clearly described in the literature and at least three different interpretations have been proposed: N-H bond [35,36,37], complex O-Ti-N compounds and compounds with nitrogen by type of embedding. However, comparing the spectra of Ti2p (Figure 4a), C1s (Figure 4c), and O1s (Figure 4b) with the spectrum of N1s (Figure 4d), some features can be distinguished. First, a reduction for the relative peak intensity for the C1s spectrum (Figure 4c) in the region of 289.2 eV indicates a degradation for C=O bonds [38]. Second, plasma modification results in an intense peak near ~529.7 eV, indicating complex Ti-O-N compounds, as well as the presence of a similar type of compound in the Ti2p spectra. Thus, the peak at the binding energy near ~399.6 eV proves existence of Ti-O-N compounds. Furthermore, the presence of a shoulder in the region of 402–400 eV with a maximum at ~401.8 eV indicates the presence of N-O bonds. Nitrogen oxide compounds with different chemical composition are formed on the PLLA scaffold fibers on the surface in the absence of reactive oxygen. This process occurs due to the plasma during sputtering that stimulates the destruction of the C-O compound of the polymer surface fibers and due to the additional interaction of chemically active nitrogen. It should be noted that the addition of xenon changes both the relative intensity of this peak and its shift to the region of lower binding energies by ~0.3 eV. This indicates an increase in the electron density of the O atoms comprising the N-O compound and a change in the stoichiometric composition of the Ti-O-N coatings formed. An intense peak is also observed in the spectrum of N1s scaffolds surface-modified in a pure xenon atmosphere, which is centered at the binding energy of 400.8 eV. The presence of this peak is due to molecularly chemisorbed nitrogen forming stable chemical bonds with the PLLA polymer backbone after the process of plasma modification [39,40]. It should be noted that the formation mechanism of nitrogenous titanium oxide coatings with alternating chemical composition is described in more detail in reference [8].

### 3.4. Roughness of the PLLA Scaffolds

Figure 5 shows the RMS roughness values obtained at different scanning areas.

The root mean square (RMS) roughness of the sample surfaces was obtained by the processing of atomic force microscopy images (AFM) using the Gwyddion software. As a result, the RMS for the area of 40 × 40 μm^2^ is in the range of (1.5 ± 0.4) µm (Figure 5 and Appendix A) and significantly independent (*p* < 0.05) of the surface modification process and mixing ratios of nitrogen (N_2_) and xenon (Xe). It should be noted that roughness in large areas depends on various factors, such as chaotic distribution of fibers, manufacturing and coating processes. Therefore, no significant differences in the roughness at 40 × 40 μm^2^ were found between the scaffold samples. The RMS roughness measured of the unmodified control PLLA scaffold measured at 3 × 3 μm^2^ (Figure 5b) has been found to be (16.7 ± 4.6) nm. In the case of PLLA scaffold surface-modified using a pristine nitrogen atmosphere, the roughness decreases to (0.9 ± 0.3) nm, and the maximum after the process of plasma modification is the roughness value for the scaffold samples N_2_ 50% + Xe 50%. No reliable differences in roughness values are observed for the studied scaffold samples Xe 100% and N_2_ 25% + Xe 75%. The decrease of roughness value after the process of surface modification by plasma treatment is caused by temperature and radiation effects (e.g., plasma etching) of the plasma [41]. In this case, plasma etching is most effective for pronounced surface topography (convexities, protruding fibers). On the other hand, there is a dependence of roughness on plasma composition and mixing ratio of plasma-forming gases [42]. Thus, the minimal surface roughness of scaffold samples surface-modified in pure N_2_ is due to the high fraction of high-energy ionized N^2+^ cations reflected from the target, which causing the plasma etching effect on scaffold surfaces [43]. An opposite trend has been observed on PLLA scaffolds surface-modified with pure xenon, where the roughness values are maximum, which is due to the prevailing target bombardment processes. This fact is also consistent with the results of the thickness measurement, which are highest for the scaffold samples surface-modified using pure xenon. AFM micrographs of the investigated scanning areas are shown in Appendix A.

### 3.5. Cell Adhesion, Proliferative Activity and Gene Expression

The level cell adherence as well as proliferative activity exhibited by HOS is shown in Figure 6.

Investigation of cell adhesion indicates that on day 1 on all PLLA scaffold samples, except the N_2_ 100% samples, cells form a dense two-dimensional layer, there is almost 80–90% colonization of the entire surface of the PLLA scaffolds (Figure 6a), but the formation of large islets is not observed. On the fifth day of incubation, cells colonize the entire surface area of all PLLA scaffold samples examined. Due to the high motility, the cells form small islets, while counting of the cells shows no statistically significant difference in their number both relative to the unmodified control samples and between the surface-modified samples. All findings of the cell adhesion studies performed upon 1 day and 5 days with incubation compared with non-modified scaffold samples can be seen from Figure 6a as well as Appendix A. Noteworthy is also the weak effect of the surface roughness of the scaffold on the adhesion of HOS cells. The increase in cell adhesion on a less rough surface is likely due to the formation of a thin, nitrogenous titanium layer on the scaffold surface. This effect of an increased cell adhesion on a less rough surface has been known and has been described in other work [44,45]. Moreover, an increase in cell adhesion with an increase in nitrogen concentration has also been observed and reported [46,47].

The Figure 6b shows that after the first day of cell culture, the proliferative activity values reliably increased only for the N_2_ 50% + Xe 50% scaffold samples, and no significant differences were observed for the other groups compared with the unmodified control samples. On the third day of HOS cell culture, there is a slight increase in the optical density of the solution, but no significant differences can be observed between the samples. On the fifth day, the values of the optical densities of the solutions are the same for all the investigated scaffold samples, but slightly lower than it was on the third day of HOS cell cultivation. This fact is related to the phenomenon that the amount of HOS cells present on the PLLA scaffold surfaces began to exceed the area they could occupy by growth and the nutrient medium was already being depleted, leading to a decrease in the proliferative activity [48].

Thus, the results of the MTT assay (Figure 6b) confirm the proliferative activity data—all PLLA scaffold samples examined have no cytotoxic towards HOS cells, and the cells adhere and proliferate well. In addition, several points should be highlighted that relate the chemical composition of scaffold fiber surfaces and their roughness to cell adhesion and proliferative activity of HOS cells. It is known that nitrogen oxide (NO) compounds can stimulate the cell proliferative activity and differentiation of HOS in the osteogenic direction [49,50], as well as titanium oxide (Ti_x_O_y_) compounds, and titanium coatings have low cytotoxicity [51]. EDX mapping micrographs of titanium compared to cell adhesion micrographs of the HOS cells on the scaffolds sample surfaces and their overlay confirm this point (Appendix A). Thus, the biological properties of nitrogenous titanium coatings on PLLA scaffold fibers depend on their stoichiometric content for Ti, N, and O. In fact, the results of the XPS analysis (Figure 4) show that the coatings on PLLA scaffolds prepared in a gas mixture of N_2_ 50% + Xe 50% exhibit the greatest proliferative activity and have a titanium to nitrogen ratio almost 1:1. Similarly, the highest titanium concentration observed for coatings prepared in pure xenon inhibits cell adhesion during the first day of HOS cell culture. On the other hand, the highest nitrogen concentration observed in the coatings prepared in the gas mixture of N_2_ 50% + Xe 50% demonstrates the highest cell adhesion. No significant differences were observed in the adhesion and proliferative activity of the HOS cell line for the other scaffold samples. The lack of reliable differences between the scaffold samples with different mixing ratios of xenon and nitrogen may not seem very impressive. However, these results represent a convincing observation that confirms the main result of this work—via plasma treatment, surface-modified PLLA scaffolds are a suitable substrate for cell adhesion and proliferative activity of HOS cells. These results are in agreement with reference [52].

It is known that HOS cells have the ability to differentiate into osteoblasts [53]. At the same time, the most effective markers of their osteogenic differentiation are osteocalcin (BGLAP) and osteopathin (SPP1). Figure 7 shows the phenotypic expression data of BGLAP and SPP1. Since BGLAP and SPP1 are late osteogenic markers, data were assessed after 14 days of cultivation of HOS on the scaffold sample surfaces [54].

Examination of osteoinductive properties using the BGLAP and SPP1 genes (Figure 7) shows that increased stimulation of osteopanthin and osteocalcin gene expression is present for all PLLA scaffold samples at day 14 after the start of incubation, indicating directional differentiation of the cells in the osteogenic direction (Figure 7). At the same time, there are significant differences both in comparison with the control samples (the nutrient medium) and between the scaffold samples. Thus, the unmodified control and N_2_ 100% samples have significantly higher expression of BGLAP and SPP1 genes than the scaffold samples N_2_ 25% + Xe 75%, N_2_ 50% + Xe 50% and Xe 100%. These results are due to the following factors. Nitric oxide (NO) molecules are known to be among the most abundant neurotransmitters in the body and act mainly as regulators of blood pressure and blood flow by inducing tonic relaxation of arterial muscles [55]. As a result of further studies, a number of other functions in cell signaling were added to the regulatory functions of blood flow. Thus, NO was found to be involved in the osteogenic response of bone to mechanical stimulation. In reference [56], it has been shown that cNOS (endothelial NO synthase) is present in osteoblastic cells and stimulates the proliferative activity of osteoblasts in vitro at low concentrations, whereas high concentrations of NO suppress their proliferative activity [57,58]. It is reasonably that the relative concentration of NO in the coatings obtained in pure N_2_ atmosphere provides the most favorable environment for osteoinduction of BGLAP and SPP1 genes by HOS cells. On the other hand, the absence of effect of scaffold surface roughness for the P–Control group on gene expression is worth mentioning. It is known that increasing the surface roughness in the submicron range can enhance the osteogenic capacity in vitro [59]. Furthermore, reference [60] shows that the expression of bone morphogenetic protein (BMP-2) by macrophages grown on a titanium surface with a granular surface was significantly higher than on a polished surface.

## 4. Conclusions

This study demonstrates the feasibility of employing a gas mixture of nitrogen and xenon in various mixing ratios during direct current magnetron sputtering of a titanium target to modify the fiber surfaces of bioresorbable poly-L-lactic acid scaffolds prepared by electrospinning. Poly-L-lactic acid scaffolds have an average fiber diameter of 1.7 ± 0.4 μm, which does not significantly change during the surface modification process and does not depend on the mixing ratios of the working gases. At the same time, a decrease of the glycerol contact angle up to 32° ± 1° has been observed for the samples surface-modified in the mixing ratio of 50% nitrogen and 50% xenon. The findings obtained via X-ray photoelectron spectroscopy indicate that the coating composition on the surface of poly-L-lactic acid scaffold fibers is a complex composition of titanium oxides and titanium oxynitrides, the ratio as well as the stoichiometric composition slightly depend on the used mixing ratio of nitrogen and xenon. For the samples prepared with 50% nitrogen and 50% xenon, the lowest coating thickness of (1.1 ± 0.3) nm and the highest roughness value of (8.1 ± 1.9) nm (for the non-modified PLLA scaffold (16.7 ± 4.6) nm) were observed. The decrease in the roughness after the process of surface modification by plasma treatment is due to temperature and radiation effects from the plasma. In vitro experiments on HOS cells show that the scaffold samples evaluated have no cytotoxic effect towards HOS cells. The cells adhere and proliferate well on each of the surface-modified scaffold samples, stimulating cell differentiation in the osteogenic direction. However, based on the obtained physicochemical and biomedical results, the most suitable surface modification of poly-L-lactic acid scaffolds for tissue engineering applications is in a gas mixture of 50% nitrogen and 50% xenon. The obtained results extend the ideas about the influence of magnetron discharge plasma, generated during reactive magnetron sputtering of a titanium target in an atmosphere of nitrogen and xenon at different mixing ratios, on the physicochemical and biomedical properties of polylactic acid scaffolds. However, polylactic acid does not have sufficient stiffness and strength to play a supporting role in the regeneration of bone tissue, as is the case with titanium implants, for example. Their use is use in bone tissue regeneration is therefore limited to mimicking the extracellular matrix for cell culturing when filling bone tissue defects. In this way, the scaffold porosity has an enormous influence on the regeneration of cancellous bone (internal bone substance, >90% porosity) [1]. In summary, these poly-L-lactic acid scaffolds are suitable as tissue-engineered structures and systems for therapy, rehabilitation and restoration of lost functions in bone tissue reconstruction due to birth defects, diseases, aging signs and injuries.

## Figures and Tables

**Figure 1 polymers-15-02969-f001:**
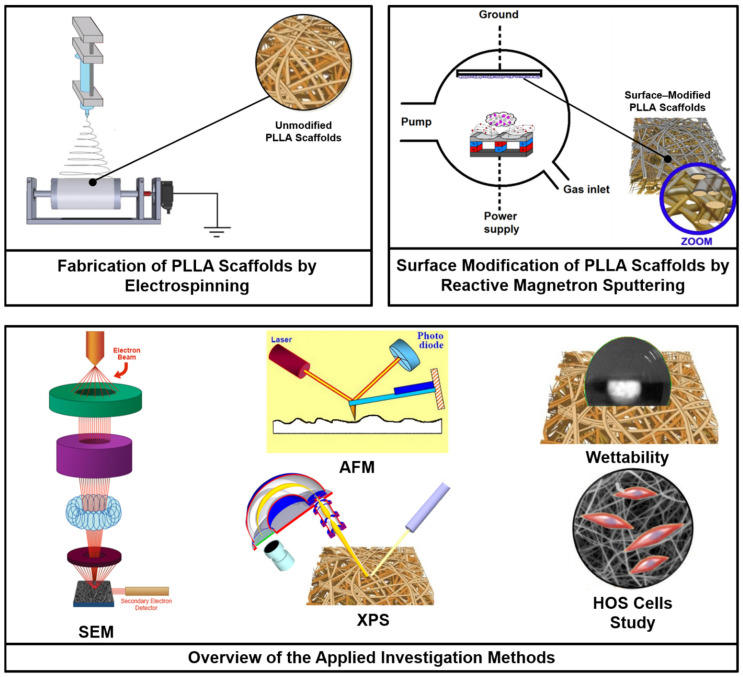
Experimental scheme of the fabrication of PLLA scaffolds, their surface modification via reactive DC magnetron sputtering of a titanium target using a gas mixture of nitrogen and xenon at different mixing ratios, and the investigation methods applied in this study.

**Figure 2 polymers-15-02969-f002:**
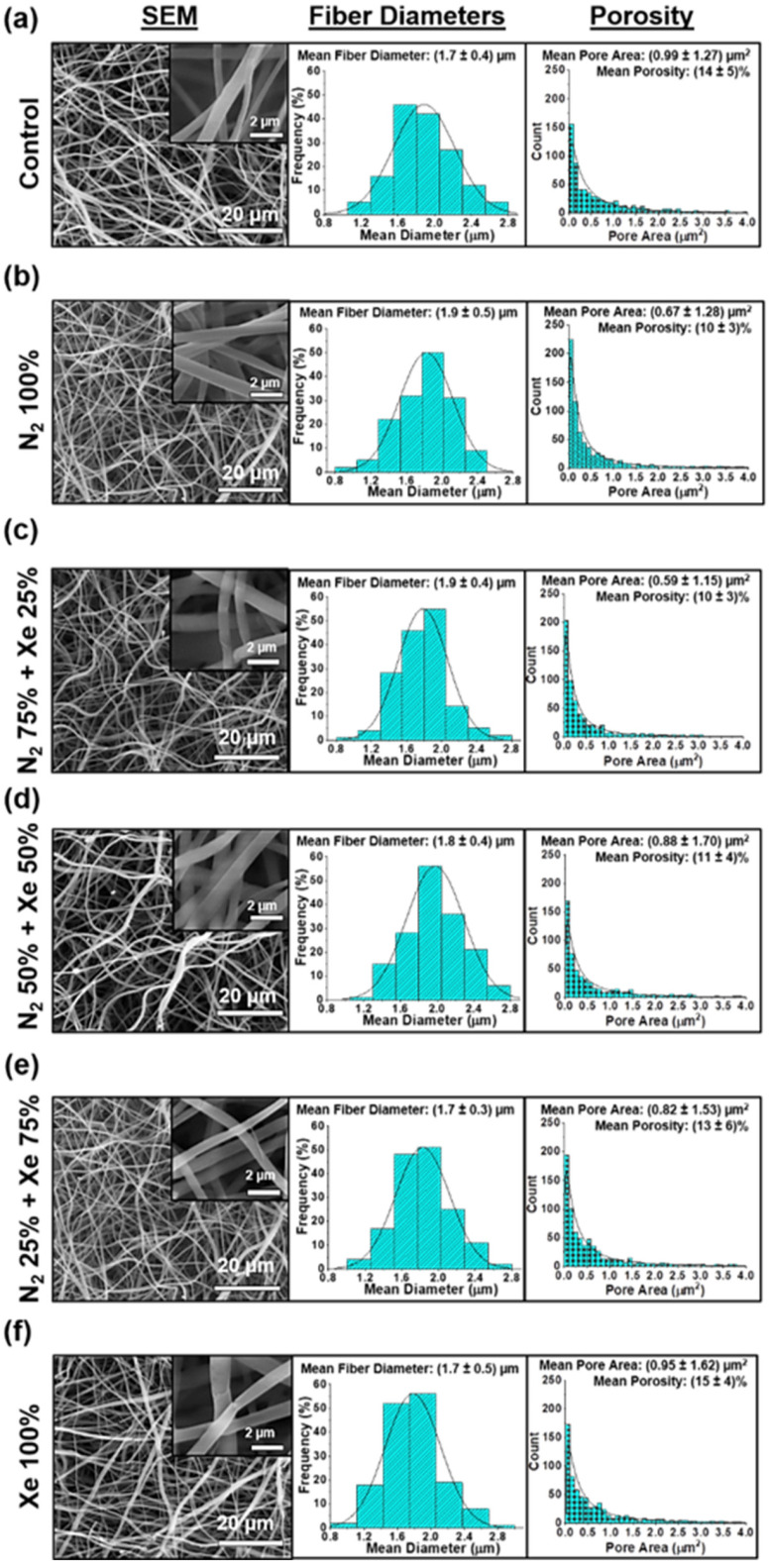
Scanning electron microscope (SEM) micrographs on the left, histograms of the fiber diameters of the scaffold samples in the middle, and histograms of the pore areas of the scaffolds on the right: (**a**) Control samples (PLLA scaffolds without surface modification), (**b**) N_2_ 100%, (**c**) N_2_ 75% + Xe 25%, (**d**) N_2_ 50% + Xe 50%, (**e**) N_2_ 25% + Xe 75%, (**f**) Xe 100%.

**Figure 3 polymers-15-02969-f003:**
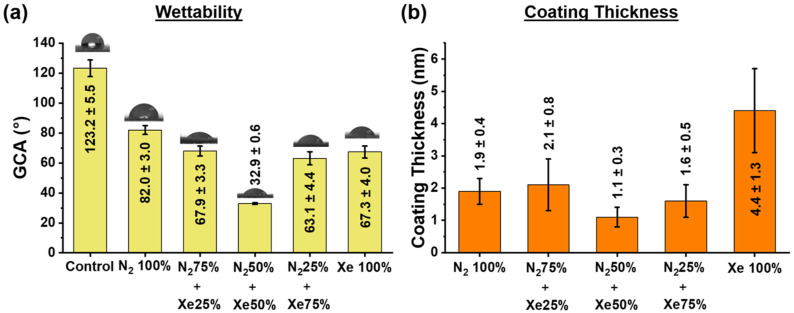
(**a**) Glycerol contact angle (GCA) for the PLLA scaffold samples under investigation, and (**b**) coating thicknesses of all surface-modified samples.

**Figure 4 polymers-15-02969-f004:**
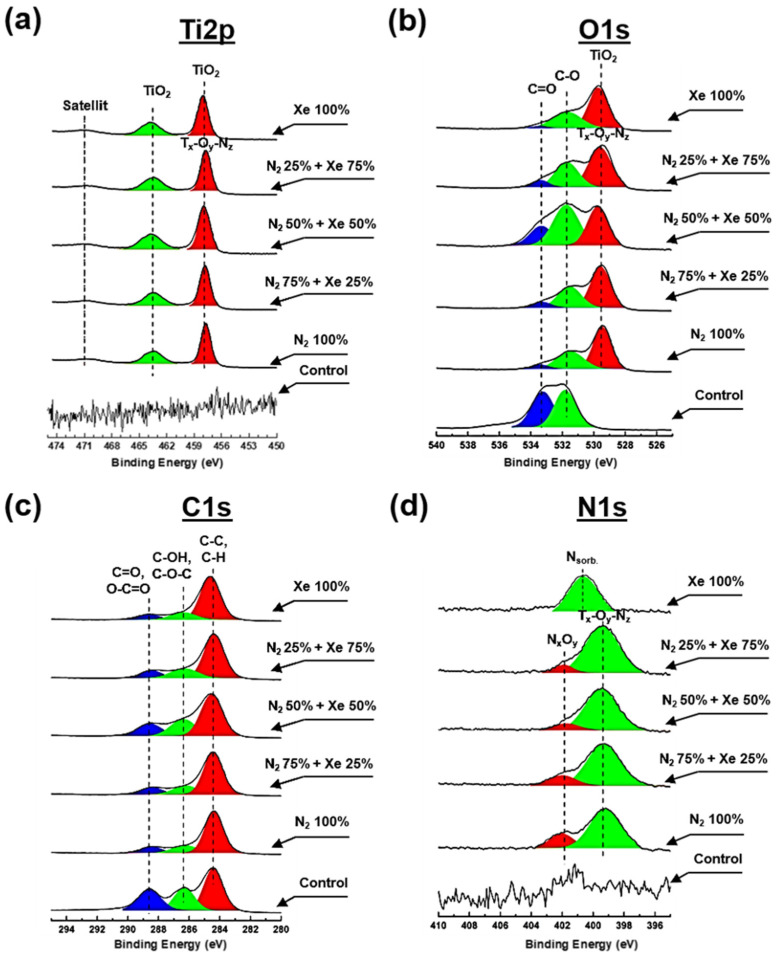
High-resolution spectra obtained via XPS for all PLLA scaffold samples that are surface-modified with a mixture of nitrogen and xenon at mixing ratios: (**a**) Ti2p, (**b**) O1s, (**c**) N1s and (**d**) C1s.

**Figure 5 polymers-15-02969-f005:**
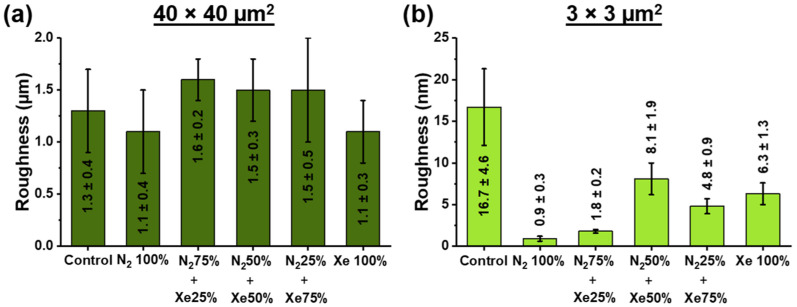
Root mean square (RMS) roughness obtained by atomic force microscopy (AFM) at two different sized scanning areas of unmodified and surface-modified PLLA scaffold samples obtained at different mixing ratios of the working gases nitrogen and xenon at (**a**) 40 × 40 μm^2^ scanning area and (**b**) 3 × 3 μm^2^ scanning area.

**Figure 6 polymers-15-02969-f006:**
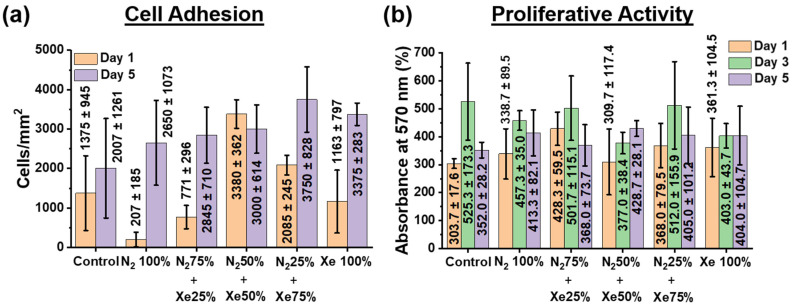
(**a**) Amount for HOS cells present at the PLLA scaffold sample surfaces surface-modified using different mixing ratios of the working gases nitrogen and xenon. (**b**) Proliferative activity of HOS cells assessed via MTT assay (measured by absorbance and presents therefore the optical density).

**Figure 7 polymers-15-02969-f007:**
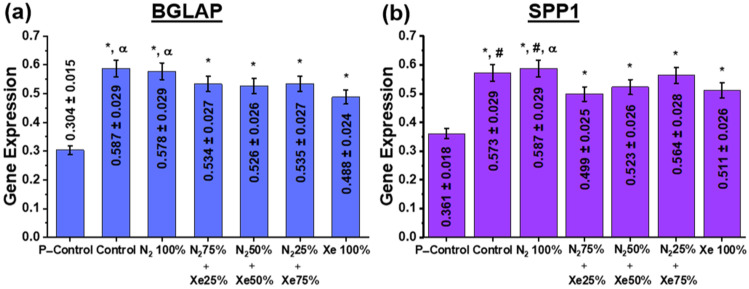
Expression of (**a**) BGLAP and (**b**) SPP1 genes in HOS cells at day 14 after the start of incubation. Here, the P-Control samples indicate the level of proliferative activity of HOS cells placed into a polymeric Petri dish and then set to 100%. The statistical legend is as follows: *—statistically significant differences to the control samples are *p* < 0.05; #—the statistically significant differences to the samples N_2_ 75% + Xe 25% are at *p* < 0.05; α—the statistically significant differences to the samples Xe 100% being *p* < 0.05.

## Data Availability

Data underlying the results presented in this paper are not publicly available at this time but may be obtained from the authors upon reasonable request.

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
