# Peer review of "Electrospun Poly-L-Lactic Acid Scaffolds Surface-Modified via Reactive Magnetron Sputtering Using Different Mixing Ratios of Nitrogen and Xenon"

_polymers, 2023, doi:10.3390/polym15132969_

Round 1
Reviewer 1 Report
Comments and suggestions
1. The rationale of the study must be mentioned in one sentence in the abstract also.
2. Line 99-103: “To study the influence of the volume concentration of nitrogen and xenon on the physicochemical and biological properties of PLLA scaffolds, polymeric fibre scaffolds were formed using a 3 wt.% electrospinning solution of PURASORB®PL-18 (average molecular weight of 217.000 – 225.000 g/mol, Corbion Purac, Amsterdam, The Netherlands) in trichloromethane (CHCl3, 97 wt.%) (EKROS, St. Petersburg, Russia) were used”, correct the sentence.
3. Use the unit of time as “h” rather than “hour” throughout the manuscript.
4. Line 124-126: “First, N2 was injected into the working chamber until the pressure was 0.4 Pa and second, and then the noble gas Xe was injected until the pressure reached 0.8 Pa”, It is not what does this statement meant for?
5. Line 138-140: The thickness and roughness of the nitrogen-containing titanium coatings on the surface of the PLLA scaffolds were measured in semicontact mode using AFM. Briefly describe the methodology with proper references.
6. Line 182-183: “Subsequently, 200 L of the RPMI medium was added to each well of the culture plate and incubated at 37 °C and 5% CO2”, what was the duration of incubation?
7. Line 250-253: “The mean pore areas and mean porosity are reliably consistent for all experimental scaffold groups and are independent of the mixing ratio of nitrogen and xenon (Figure 2, right)”, how much this statement is true? It is obvious in the Figures there are significant alterations in the mean pore area and mean porosity in the different scaffolds as compared to the control one. Justification needed for this.
8. Line 282-284: “In addition, it is known that the surface roughness also significantly affects the values of the wetting angle”, what kind of affect? Positive or negative, explanation needed at this point, which will help to the readers.
9. As mentioned that, no cytotoxic effect towards HOS cells towards the scaffold samples was observed, what about the control group used for the same experiment?
10. “Figure 7. Expression of a) BGLAP and b) SPP1 genes in HOS cells at day 14 after the start of incubation. P-Control represents the level of proliferative activity of HOS cells in a plastic Petri dish and is set to 100%”, recheck the values on x-axis of the bar diagram in Figure (b).
11. In conclusion as stated “based on the obtained physicochemical and biomedical results, the most suitable surface modification of poly-L-lactic acid scaffolds for tissue engineering applications is in a gas mixture of 50% nitrogen and 50% xenon”, how much this statement is correct and justifiable in light of the gene expression experiment on HOS cells as presented in Figure 7a?
12. No significant differences were observed in the BGLAP gene expressions in the treated groups by 75% N2 + 25% Xe, 50% N2 + 50% Xe as well as 25% N2 + 75% Xe as shown in Figure 7a, then it was concluded that 50% N2 + 50% Xe was the best product?
Minor editing of English language required
Author Response
Dear reviewer,
we thank you for your valuable comments and suggestions. We revised the manuscript according to your comments and marked the changes in green in the manuscript text. For a point-by-point response to your comments, please see below.
Sincerely,
Sven Rutkowski and Sergei I. Tverdokhlebov
Reviewer 1:
- The rationale of the study must be mentioned in one sentence in the abstract also.
Answer 1: We thank the reviewer for this comment. A sentence was added in the abstract as requested.
- Line 99-103: “To study the influence of the volume concentration of nitrogen and xenon on the physicochemical and biological properties of PLLA scaffolds, polymeric fibre scaffolds were formed using a 3 wt.% electrospinning solution of PURASORB®PL-18 (average molecular weight of 217.000 – 225.000 g/mol, Corbion Purac, Amsterdam, The Netherlands) in trichloromethane (CHCl3, 97 wt.%) (EKROS, St. Petersburg, Russia) were used”, correct the sentence.
Answer 2: We would like to thank the reviewer for pointing out this issue. This sentence has been corrected.
- Use the unit of time as “h” rather than “hour” throughout the manuscript.
Answer 3: We thank the reviewer for this comment. For hours, the "h" is now used in the manuscript.
- Line 124-126: “First, N2was injected into the working chamber until the pressure was 0.4 Pa and second, and then the noble gas Xe was injected until the pressure reached 0.8 Pa”, It is not what does this statement meant for?
Answer 4: Thanks reviewer for this question. Since the statement in this sentence is misleading and physically doubtful, this sentence has been deleted.
- Line 138-140: The thickness and roughness of the nitrogen-containing titanium coatings on the surface of the PLLA scaffolds were measured in semicontact mode using AFM. Briefly describe the methodology with proper references.
Answer 5: We thank the reviewer and are grateful to this comment. This part has been rewritten and adjusted as requested.
- Line 182-183: “Subsequently, 200 mL of the RPMI medium was added to each well of the culture plate and incubated at 37 °C and 5% CO2”, what was the duration of incubation?
Answer 6: We thank the reviewer for this question. The incubation time is given in Chapter 2 in the paragraphs for the MTT assays and for the cell adhesion tests. In the paragraph where the sentence is found, the cell seeding procedure is described. The total incubation time was 5 days (until all tests were completed). For the MTT assays, incubation was interrupted on days 1, 3, and 5 after cell seeding to perform the measurements. For micrographs for the cell adhesion tests, incubation has been interrupted on days 1 and 5 after cell seeding. For gene differentiation tests, incubation was interrupted on day 14 after cell seeding.
- Line 250-253: “The mean pore areas and mean porosity are reliably consistent for all experimental scaffold groups and are independent of the mixing ratio of nitrogen and xenon (Figure 2, right)”, how much this statement is true? It is obvious in the Figures there are significant alterations in the mean pore area and mean porosity in the different scaffolds as compared to the control one. Justification needed for this.
Answer 7: We thank the reviewer for this comment. The pore areas and mean porosities were calculated using five microscope images at different locations on the framework and statistically tested using a one-way ANOVA test. All results obtained were considered statistically significant at p > 0.05. Therefore, all results in the histogram legend have no statistically significant changes using this method. This point was added into chapter 2.
- Line 282-284: “In addition, it is known that the surface roughness also significantly affects the values of the wetting angle”, what kind of affect? Positive or negative, explanation needed at this point, which will help to the readers.
Answer 8: We thank the reviewer for this issue. We thank the reviewer for this comment. This point has been included in the manuscript.
- As mentioned that, no cytotoxic effect towards HOS cells towards the scaffold samples was observed, what about the control group used for the same experiment?
Answer 9: We would like to thank the reviewer for this question. We also ran control samples in plastic Petri dishes to make sure the cell line was growing well and all cells were alive. However, we did not show it in the results because no cytotoxicity was found here, which was to be expected. Therefore, in the results of the study, we presented the tested titanium samples as control in the manuscript.
- “Figure 7. Expression of a) BGLAP and b) SPP1 genes in HOS cells at day 14 after the start of incubation. P-Control represents the level of proliferative activity of HOS cells in a plastic Petri dish and is set to 100%”, recheck the values on x-axis of the bar diagram in Figure (b).
Answer 10: We are very grateful to the reviewer for finding this issue. This issue has been fixed, and an updated version of this figure has been included in the manuscript.
- In conclusion as stated “based on the obtained physicochemical and biomedical results, the most suitable surface modification of poly-L-lactic acid scaffolds for tissue engineering applications is in a gas mixture of 50% nitrogen and 50% xenon”, how much this statement is correct and justifiable in light of the gene expression experiment on HOS cells as presented in Figure 7a?
Answer 11: Considering that the wettability and roughness as well as the area of cell colonies (see SI Figure S2) of the experimental group with 50% nitrogen and 50% xenon were significantly higher than those of the other samples investigated, this sample group was identified as optimal.
- No significant differences were observed in the BGLAP gene expressions in the treated groups by 75% N2+ 25% Xe, 50% N2 + 50% Xe as well as 25% N2 + 75% Xe as shown in Figure 7a, then it was concluded that 50% N2 + 50% Xe was the best product?
Answer 12: We thank the reviewer for this question. As correctly noted by the reviewer, the sample groups 75% nitrogen + 25% xenon, 50% nitrogen + 50% xenon and 25% nitrogen + 75% xenon do not show significant differences in the level of gene expression. Considering that the wettability and roughness, as well as the cell colony area (see SI Figure S2) of the experimental group 50% nitrogen + 50% xenon are significantly higher than those of the other sample groups, this sample group was selected as optimal.
Reviewer 2 Report
COMMENTS FOR AUTHORS:
1. All the abbreviations must be defined when they are used first time in the manuscript, including the abstract. E.g., HOS, etc.
2. In Section 2.2: Authors should provide a flow chart of producing the a scaffold clearly. How and in which step the titanium was reacted with N2 is not clear. How the Ti was bonded with N2 is also not investigated. What is the role of Ti in the surface modification? Explain.
3. Page-4: "root mean roughness (RMS) " --Check the abbreviation or the parameter properly.
4. Page-4: Check the spelling of "diiodmethane".
5. What are the dimensions of the sample for cell culture study?
6. Page-5: "with a ×10 objective" --should be "with a ×10 objective lens".
7. What is the moto to do the Glycerol contact angle (GCA) for the PLLA scaffold samples instead of water contact angle study?
8. Page-8: "The highest coating thickness is observed for the pure Xe atmosphere, and the lowest for the N250% + Xe50% mixed gas atmosphere."--Page-9: "Since the value of the ionization cross-section of xenon is higher than that of nitrogen [22], the thickness of the coatings formed in an atmosphere of pure xenon is higher."-- In this principle, lowest thickness should come for the N275% + Xe25% mixed gas atmosphere. Justify!
9. In Figure 4a, what is "Satellit"?
10. In Figure 5b, it is clearly seen that Control has higher roughness than that of any other surface modified samples. Then, how the surface modification helped in cell adhesion and proliferation.
11. Page-11: the roughness values should be written as "16.7 ± 4.6 nm" or 0.9 ± 0.3 nm. instead of "is (16.7 ± 4.6) nm" or "(0.9 ± 0.3) nm". And specify which roughness parameter is it.
12. Page-12: How the "nitrogen oxide (NO)" is formed? What kind of compounds it forms?
13. Page-12: How the "titanium oxide (TiO)" is formed? What kind of compounds it forms?
14. The Conclusions is very general. It must be written with specific points and numerical values of some important characteristics.
15. Since these scaffolds intend to be used for bone tissue reconstruction, mechanical properties of this scaffold should be conducted otherwise, it has no use for that purpose.
16. More References should be from recent study.
Author Response
Dear reviewer,
we thank you for your valuable comments and suggestions. We revised the manuscript according to your comments and marked the changes in green in the manuscript text. For a point-by-point response to your comments, please see below.
Sincerely,
Sven Rutkowski and Sergei I. Tverdokhlebov
Reviewer 2:
- All the abbreviations must be defined when they are used first time in the manuscript, including the abstract. E.g., HOS, etc.
Answer 1: We thank the reviewer for pointing out this issue. We checked this point carefully for the manuscript and edited this point according to the requirement.
- In Section 2.2: Authors should provide a flow chart of producing the a scaffold clearly. How and in which step the titanium was reacted with N2 is not clear. How the Ti was bonded with N2 is also not investigated. What is the role of Ti in the surface modification? Explain.
Answer 2: We thank the reviewer for this question. Chemically active titanium particles are formed by bombarding a titanium target with xenon and nitrogen ions during magnetron sputtering. In this case, the chemical reaction of interaction between titanium and nitrogen occurs not in the plasma of the magnetron discharge, but on the surface of scaffolds (substrate). This process is described in our previous work [10.1007/s11090-019-09956-x] and explained theoretically in [10.1088/1742-6596/1259/1/012019]. The role of titanium for biomedical modifications concerns its biocompatibility, corrosion behavior, mechanical behavior [10.1504/IJNBM.2007.016517], and antibacterial properties [10.1016/j.msec.2015.12.062].
- Page-4: "root mean roughness (RMS) " --Check the abbreviation or the parameter properly.
Answer 3: We thank the reviewer a lot for finding this issue. We are sorry for this mistake. This issue has been fixed in the manuscript text.
- Page-4: Check the spelling of "diiodmethane".
Answer 4: We thank the reviewer for finding this spelling mistake. This problem has been fixed.
- What are the dimensions of the sample for cell culture study?
Answer 5: We thank the reviewer for this question. The samples for the cell studies were circular and had an area of 1 cm2. This information has been added to the manuscript in the corresponding paragraphs in Chapter 2.
- Page-5: "with a ×10 objective" --should be "with a ×10 objective lens".
Answer 6: We thank the reviewer for pointing this out. This point has been adjusted in the manuscript.
- What is the moto to do the Glycerol contact angle (GCA) for the PLLA scaffold samples instead of water contact angle study?
Answer 7: We thank the reviewer for this very good question. Glycerol was chosen as the only wetting liquid because it is viscous enough to wet the scaffold samples, as other liquid were not suitable to be used to properly measure the contact angles of the surface-modified scaffolds. This information was added to Chapter 2.3 of the manuscript.
- Page-8: "The highest coating thickness is observed for the pure Xe atmosphere, and the lowest for the N250% + Xe50% mixed gas atmosphere."--Page-9: "Since the value of the ionization cross-section of xenon is higher than that of nitrogen [22], the thickness of the coatings formed in an atmosphere of pure xenon is higher."-- In this principle, lowest thickness should come for the N275% + Xe25% mixed gas atmosphere. Justify!
Answer 8: We thank the reviewer for his careful reading of the manuscript. Indeed, the larger the ionization cross-section, the higher the probability of generating a chemically active ion and the higher the sputtering rate. However, in the case of a complex working gas composition, the interaction process is not linear and depends on the nitrogen and xenon concentrations (mixing ratios), the Penning effect, the temperature of the ions, the ion heating by wave attenuation and Coulomb scattering. Since this work is of applied nature, a more detailed study of plasmas has not been carried out. In the future, we plan to carry out a suitable research project for this purpose.
- In Figure 4a, what is "Satellit"?
Answer 9: We thank the reviewer for this question. The "satellite" peak in Figure 4a is an additional peak in the XPS spectrum indicating the presence of titanium oxide compounds in the coating and is due to the transition of the valence electron to a higher energy level [10.7567/JJAPS.32S2.113]. This point has already been stated in the paragraph under Figure 4 and extended to include this explanation.
- In Figure 5b, it is clearly seen that Control has higher roughness than that of any other surface modified samples. Then, how the surface modification helped in cell adhesion and proliferation.
Answer 10: We thank the reviewer for this comment. The increase in cell adhesion on a less rough surface is likely due to a decrease in the glycerol contact angle and the formation of a thin, nitrogenous titanium layer on the scaffold surface. Moreover, the effect of increased cell adhesion on a less rough surface was reported by the authors of the following studies [10.1002/(SICI)1097-4636(199707)36:1<99::AID-JBM12>3.0.CO;2-E, 10.1002/jbm.820290314]. It should be also noted that the authors of these studies [10.1016/j.surfcoat.2006.03.051, 10.1016/j.surfcoat.2010.12.044] also observed an increase in cell adhesion with an increase in nitrogen concentration.
- Page-11: the roughness values should be written as "16.7 ± 4.6 nm" or 0.9 ± 0.3 nm. instead of "is (16.7 ± 4.6) nm" or "(0.9 ± 0.3) nm". And specify which roughness parameter is it.
Answer 11: We thank the reviewer for this comment. These points were revised as suggested.
- Page-12: How the "nitrogen oxide (NO)" is formed? What kind of compounds it forms?
Answer 12: We thank the reviewer for these questions. Nitric oxide compounds of changing chemical composition formation on the PLLA of the scaffolds surface in the absence of reactive oxygen, is due to the processes of plasma surface destruction of the C-O compound of the polymer surface fibers and the interaction of chemically active species of nitrogen. This mechanism was studied in detail in a previous work [10.1007/s11090-019-09956-x].
- Page-12: How the "titanium oxide (TiO)" is formed? What kind of compounds it forms?
Answer 13: We thank the reviewer for these questions. The formation of titanium oxide (TixOy) and titanium oxynitride (TiOxNy) containing compounds on the PLLA surface of the scaffolds in the absence of reactive oxygen is due to the destruction of the C-O bond of the outer polymer fibers by the plasma. This mechanism was studied in more details in a previous work [10.1007/s11090-019-09956-x].
- The Conclusions is very general. It must be written with specific points and numerical values of some important characteristics.
Answer 14: We thank the reviewer for this remark. The conclusion has been adjusted as requested.
- Since these scaffolds intend to be used for bone tissue reconstruction, mechanical properties of this scaffold should be conducted otherwise, it has no use for that purpose.
Answer 15: We thank the reviewer for this comment. Certainly, the mechanical properties of scaffolds are an important characteristic. Polymeric polylactic acid (PLA) frameworks do not have the stiffness and strength that titanium implants do, for example. Their use in bone tissue regeneration is limited to mimicking the extracellular matrix (ECM) for cell culturing when filling bone defects. For example, the authors of the review [10.1016/j.jsamd.2020.01.007] show a tremendous influence of porosity on the regenerative potential of the scaffold in the restoration of cancellous bone. Therefore, our studies with PLA are related to the improvements of surface coatings to further enhance cell cultivation.
- More References should be from recent study.
Answer 16: We thank the reviewer for this comment. The difficulty in selecting relevant recent scientific literature is due to the lack of studies on the effects of the mixing ratio of nitrogen and xenon on the properties of polymeric bioresorbable scaffolds. Therefore, older but cited literature from major scientific publications was used to explain the identified dependencies. We hope that this fact does not diminish the relevance of our study. However, as we are aware of the relevance of more recent studies for citation in new scientific papers, we have tried to replace some older studies with more recent ones where possible.
Round 2
Reviewer 1 Report
Authors have revised the manuscript as per the suggestions and comments raised.
Author Response
We thank the reviewer for this judgement and the revision of our work once again.